# Carbonic Anhydrase 2 Deletion Delays the Growth of Kidney Cysts Whereas Foxi1 Deletion Completely Abrogates Cystogenesis in TSC

**DOI:** 10.3390/ijms25094772

**Published:** 2024-04-27

**Authors:** Sharon Barone, Kamyar Zahedi, Marybeth Brooks, Manoocher Soleimani

**Affiliations:** 1Research Services, New Mexico Veterans Health Care System, Albuquerque, NM 87108, USA; sbarone@salud.unm.edu (S.B.); kzahedi@salud.unm.edu (K.Z.); marbrooks@salud.unm.edu (M.B.); 2Department of Medicine, Division of Nephrology, University of New Mexico Health Sciences Center, Albuquerque, NM 87131, USA

**Keywords:** carbonic anhydrase 2, *Foxi1*, tuberous sclerosis, cystogenesis, mTORC1, H^+^-ATPase, A-intercalated cells

## Abstract

Tuberous sclerosis complex (TSC) presents with renal cysts and benign tumors, which eventually lead to kidney failure. The factors promoting kidney cyst formation in TSC are poorly understood. Inactivation of carbonic anhydrase 2 (*Car2)* significantly reduced, whereas, deletion of Foxi1 completely abrogated the cyst burden in *Tsc1* KO mice. In these studies, we contrasted the ontogeny of cyst burden in *Tsc1/Car2* dKO mice vs. *Tsc1/Foxi1* dKO mice. Compared to *Tsc1* KO, the *Tsc1/Car2* dKO mice showed few small cysts at 47 days of age. However, by 110 days, the kidneys showed frequent and large cysts with overwhelming numbers of A-intercalated cells in their linings. The magnitude of cyst burden in *Tsc1/Car2* dKO mice correlated with the expression levels of Foxi1 and was proportional to mTORC1 activation. This is in stark contrast to *Tsc1/Foxi1* dKO mice, which showed a remarkable absence of kidney cysts at both 47 and 110 days of age. RNA-seq data pointed to profound upregulation of *Foxi1* and kidney-collecting duct-specific H^+^-ATPase subunits in 110-day-old *Tsc1/Car2* dKO mice. We conclude that *Car2* inactivation temporarily decreases the kidney cyst burden in *Tsc1* KO mice but the cysts increase with advancing age, along with enhanced Foxi1 expression.

## 1. Introduction

Tuberous sclerosis complex (TSC) is a rare autosomal dominant genetic disease that affects over two million individuals worldwide [1,2]. It is a multi-system disease that is caused by mutations in either the *Tsc1* or *Tsc2* genes and damages various organs such as the kidneys, lungs, and brain [3]. In the kidneys, TSC presents with the enlargement of benign tumors (angiomyolipomata) and cysts, which eventually leads to kidney failure [3,4,5,6]. Although the genetic basis of TSC disease is well worked out, the factors that promote cyst formation and tumor growth in TSC are poorly understood. Previous investigations examining various mouse models of TSC have revealed that the epithelia of renal cysts maintain the integrity of their *Tsc* loci and that the loss of heterozygosity was observed in only a small number of cystic epithelial cells [7,8,9,10]. These observations are similar to those in human TSC renal cystic disease, where cells lining the cysts express both Tsc1 and Tsc2 proteins [11].

The unregulated activation of mammalian target of rapamycin complex 1 (mTORC1) is the primary factor that promotes cell proliferation, benign tumor (angiomyolipomata) development, and cystogenesis in TSC kidneys [12,13,14]. mTORC1 is a serine/threonine protein kinase complex that regulates cell growth in response to environmental factors, and its dysregulation contributes to many pathophysiologic states [15,16,17]. The activity of mTORC1 is mediated through the phosphorylation of its substrates, including the ribosomal protein S6K and eIF4E-binding proteins (4-EBP) [18].

A number of studies indicate the preponderance of A-intercalated (A-IC) cells in kidney cyst epithelia in mouse models of TSC, as well as in TSC patients [7,9,10,11]. The latter was verified by the expression of apical H^+^-ATPase, and the basolateral Cl^−^/HCO_3_^−^ exchanger AE1 (SLC4A1) via immunofluorescence labeling [7,9,10,11]. The increased expression of H^+^-ATPase in the cystic epithelium is of great interest since it is involved in the activation of mTORC1 and cell proliferation [19,20,21,22], and is most likely mediated via amino acid-sensitive interactions with Ragulator, a scaffolding complex that anchors Rag GTPases to the lysosome [19,20,21,22]. 

H^+^-ATPase is a multi-subunit complex composed of V0 (membrane-spanning) and V1 (catalytic) complexes that couple the energy of ATP hydrolysis to H^+^ translocation across the plasma and intracellular membranes [23,24]. The inward translocation of H^+^ causes the acidification of the intracellular compartments of secretory vesicles, endosomes, and lysosomes [23,24]. Chemical inhibition of H^+^-ATPase alkalinizes the pH of the lysosomal lumen and inactivates mTORC1 [25]. At the plasma membrane of A-IC cells, H^+^-ATPase is responsible for pumping H^+^ into the lumen of the collecting duct, thus regulating the systemic acid/base balance [26,27,28]. 

RNA sequencing (RNA-seq) and confirmatory expression studies in our laboratories demonstrated robust expression of Forkhead box I1 (Foxi1) transcription factor and its downstream targets, including apical H^+^-ATPase transmembrane (V0) and catalytic (V1) components, in the cyst epithelia of *Tsc1* as well as *Tsc2* knockout mice, but not in mice with the *Pkd1* gene mutation [9]. 

The electrogenic 2Cl^−^/H^+^ exchanger, CLC-5, is significantly up-regulated and shows remarkable co-localization with H^+^-ATPase on the apical membrane of cyst epithelia in various TSC mouse models, but not in *Pkd1* mutant mice [29]. The deletion of Foxi1, a protein that is vital to H^+^-ATPase expression and IC cell viability, completely inhibited mTORC1 activation and abrogated the cyst burden in 47-day-old *Tsc1/Foxi1* dKO mice [9]. These results unequivocally demonstrate the critical role that Foxi1 and IC cells, along with H^+^-ATPase, play in kidney cystogenesis in TSC. 

In this present study, we sought to examine the ontogeny of kidney cysts in *Tsc1/Car2* dKO and *Tsc1/Foxi1* dKO mice. *Tsc1/Car2* dKO animals similar to *Tsc1/Foxi1* dKO mice show either very few or no cysts compared to age-matched *Tsc1* KO mice [9]. However, unlike the *Tsc1/Foxi1* dKO mice, the *Tsc1/Car2* dKO animals develop severe renal cystic disease as they age [9,10]. Compared to *Tsc1* KO mice, which usually die before the age of 55 days, both *Tsc1/Car2* dKO and *Tsc1/Foxi1* dKO mice survive [9]. 

## 2. Results

### 2.1. Effect of Car2 Deficiency on Renal Cystogenesis in Tsc1 KO Mice

The number of A-IC cells decreases significantly in the kidneys of *Car2*-deficient mice [30,31]. This was confirmed by significant reductions in the expression levels of markers of both A-IC and B-IC cells (e.g., SLC26A4, SLC26A7, and H^+^-ATPase) [30,31,32]. In contrast, the ablation of *Tsc1* in the principal cells (PCs) of the renal-collecting duct leads to substantial expansion of A-IC cells in the epithelium of renal cysts [9,10]. Based on the above, we generated *Tsc1/Car2* dKO mice, which showed a significant reduction in the cyst burden at 47 days of age [9]. In this present study, we examined the impact of aging on the kidney cyst burden in *Tsc1/Car2* dKO mice and included age-matched *Tsc1/Foxi1* dKO mice. The histology of kidneys of WT, *Tsc1/Car2* dKO, *Tsc1/Foxi1* dKO, and *Tsc1* KO mice at 47 days of age are shown in Figure 1A–D. The results were further compared to *Tsc1/Foxi1* dKO mice at 110 days of age, as shown in Figure 1F. Our results indicate that the kidney cysts were significantly smaller and less frequent in 47-day-old *Tsc1/Car2* dKO mice compared to age-matched *Tsc1* KO mice. The *Tsc1/Car2* dKO mice displayed significant cyst burden at 110 days of age when compared to 47-day-old *Tsc1* KO mice (Figure 1B,D). In contrast, the *Tsc1/Foxi1* dKO mice displayed a complete absence of cyst formation at both 47 and 110 days of age (Figure 1C,F). The life span of *Tsc1* KO mice, which is less than 60 days, did not allow for direct comparison of cyst burden to that of 110-day-old *Tsc1/Car2* dKO mice.

### 2.2. Comparison of the Markers of A-IC Cells and Principal Cells (PCs) in the Kidneys of Tsc1 KO and Tsc1/Car2 dKO Mice

The examination of renal cysts in *Tsc1* KO mice revealed the predominance of proliferatively active A-IC cells in the cystic epithelium [9,10]. To determine if such an imbalance in favor of A-IC cells also occurs in the kidneys of *Tsc1/Car2* dKO mice, we examined the renal expression and localization of A-IC cells and PC markers, H^+^-ATPase, and Aquaporin 2 (AQP2), respectively. The results of double immunofluorescence labeling with H^+^-ATPase and AQP2 antibodies in WT, *Tsc1* KO, and *Tsc1/Car2* dKO mice at 47 days and *Tsc1/Car2* dKO mice at 110 days of age are depicted in Figure 2. The H^+^-ATPase and AQP2 double-labeling images in WT mice are depicted in Figure 2A–C. The tubular epithelium of renal cysts in *Tsc1* KO mice showed a predominant and widespread labeling with H^+^-ATPase on their apical membrane and a paucity of AQP2 labeled cells (Figure 2D–F). The labeling at 47 days of age in *Tsc1/Car2* dKO mice showed few nascent cysts, which, compared to WT collecting ducts, had a proportion of H^+^-ATPase staining cells (Figure 2G–I). The labeling in *Tsc1/Car2* dKO mice at 110 days of age is shown in Figure 2J–L and demonstrates an almost universal labeling with H^+^-ATPase on the apical membrane of kidney cysts with a few cells that show basolateral staining with AQP2. 

### 2.3. The Effect of Car2 Gene Ablation on the Activation of mTORC1 in Tsc1 KO Mice

The activation of mTORC1 is paramount to the pathology of the disease in TSC [1,14,33,34]. Therefore, we compared the activation of mTORC1 in the kidneys of WT, *Tsc1/Car2* dKO, *Tsc1/Foxi1* dKO, and *Tsc1* KO mice (Figure 3). Phosphorylation of S6 was used as a measure of mTORC1 activation. The examination of kidneys of 47 and 110-day-old *Tsc1/Car2* dKO mice revealed enhanced S6 kinase staining in *Tsc1/Car2* dKO mice compared to those of WT mice (Figure 3A,B,E). Our results further indicate that the activation of mTORC1 becomes more widespread in the kidneys of 110-day-old *Tsc1/Car2* dKO mice as the process of cystogenesis progresses (Figure 3E) when compared to *Tsc1* KO mice (Figure 3D). 

An additional comparison of kidneys from time-matched *Tsc1/Foxi1* dKO and *Tsc1/Car2* dKO mice for phospho-S6 levels revealed that while phospho-S6 staining remains comparably low in *Tsc1/Foxi1* dKO mice at both 47 and 110 days of age, it showed a wider staining pattern in 110-day-old *Tsc1/Car2* dKO mice (Figure 3C,E,F). These results confirm that the process of cystogenesis and associated hyperproliferation of A-IC cells in lining the cyst is significantly delayed when the *Car2* gene is ablated in *Tsc1* KO mice. 

### 2.4. Changes in the Transcriptome of Tsc1/Car2 dKO Mice at 47 and 110 Days of Age 

Our previous studies indicated that the process of cystogenesis in various mouse models of TSC is driven by the hyperproliferation of A-IC cells [9,10]. To determine the role of the reduction of A-IC expansion in the delayed cystogenic process in *Tsc1/Car2* dKO mice, we compared the RNA-seq analysis results of WT mice to those of 47 and 110-day-old *Tsc1/Car2* dKO mice. Comparison of WT and *Tsc1/Car2* dKO mice gene expression profiles on days 47 and 110 (Figure 4 and Figure 5; Appendix A) revealed that while the expression of A-IC cell-specific genes, such as *Foxi1*, *Slc4a1*, and certain subunits of H^+^-ATPase (e.g., *Atp6v0d2*, *Atp6v1b1*, and *Atp6v1g3*), are upregulated in the kidneys of both 47 and 110-day-old *Tsc1/Car2* dKO mice, the magnitude of expression of Foxi1 along with several H^+^-ATPase subunits (e.g., *Atp6v0a1*, *Atp6v0e*, *Atp6v1c2*, *Atp6v1d*, and *Atp6v1e1*) are robustly elevated in the kidneys of 110-day-old *Tsc1/Car2* dKO mice vs. 47-day-old *Tsc1/Car2* dKO mice (Figure 6; Appendix A). The comparison of the renal transcriptome of *Car2* KO mice to those of the 47 and 110-day-old *Tsc1/Car2* dKO mice (Appendix A) also revealed an enhanced expression of transcripts that code for *Foxi1*, *Slc4a1*, and A-IC cell-associated H^+^-ATPase subunits (Figure 4 and Figure 5). In addition, the expression of *Car12* and *Car13* mRNA levels were significantly upregulated in 110-day-old *Tsc1/Car2* dKO compared to time-matched *Car2* KO mice. The most cogent point of these results is the progressive increase in the expression of A-IC cell-specific genes in the kidneys of *Tsc1/Car2* dKO mice, as their cyst burden increases over time (day 47 vs. day 110 samples).

### 2.5. Enrichment Analysis of RNA-Seq Results 

The results of RNA-seq studies were subjected to KEGG enrichment analysis, which examined differentially expressed transcripts (DET) and revealed the presence of 79, 156, and 23 enriched terms with a false discovery rate (FDR) of less than 0.05 for 47-day-old *Tsc1/Car2* dKO vs. WT mice, 110-day-old *Tsc1/Car2* dKO vs. WT mice, and 110-day-old dKO vs. 47-day-old dKO mice, respectively (Figure 7; Appendix A). The enriched terms included “collecting duct acid secretion, PI3K-AKT and MAPK signaling pathways” (Appendix A). The “cell cycle pathway” was only recognized as an enriched term in KEGG analyses of 110-day-old *Tsc1/Car2* dKO vs. WT mice and 47-day-old vs. 110-day-old *Tsc1/Car2* dKO mice (Appendix A).

Gene ontology (GO) analyses revealed 939, 1000+, and 422 biological processes (GO-BP; Figure 8; Appendix A) and 128, 222, and 26 molecular function (GO-MF; Figure 9; Appendix A) terms for 47-day-old *Tsc1/Car2* dKO vs. WT mice; 110-day-old *Tsc1/Car2* dKO vs. WT mice, and 110-day-old dKO vs. 47-day-old dKO mice, respectively. The enrichment in cell cycle-associated terms was noticeable in both the 47 and 110-day-old *Tsc1/Car2* dKO results. The GO-BP enrichment analysis revealed that the terms associated with cell cycle and autophagy were enriched in 110-day-old *Tsc1/Car2* dKO mice compared to WT mice and 47-day-old *Tsc1/Car2* dKO mice results (Figure 8; Appendix A). The GO-Molecular Function (GO-MF) analyses identified significant enrichment in terms associated with the regulation of growth factor receptor binding, kinase activity, and GTP-binding protein activity at both 47 days vs. WT and 110 days vs. WT (Figure 9; Appendix A).

## 3. Discussion

This current study investigates the ontogeny of kidney cysts in TSC by examining the role of A-IC cell expansion in this process. In this manuscript, we compare the alterations in the renal transcriptome of two mouse models of TSC cystogenesis that lead to the disappearance of A-IC cells and abrogated cyst formation (*Tsc1/Foxi1I* dKO) or significant reduction in A-IC cell numbers and delayed cystogenesis (*Tsc1/Car2* dKO) to that of WT mice.

The mice with either *Tsc1* or *Tsc2* ablation in kidney PCs, *Tsc1* inactivation in pericytes, or *Tsc2*^+/KO^ exhibit numerous cortical cysts, which are overwhelmingly composed of hyperproliferating A-IC cells [7,9,10]. A similar cell phenotype in cystic epithelium was observed in humans with TSC, and in heterozygous *Tsc2*^+/ko^ mice [9,10]. These observations point to the presence of a similar mechanism that drives the process of kidney cystogenesis in animal models of TSC, as well as in TSC patients. This cellular phenotype profoundly contrasts with kidney cysts in Autosomal Dominant Polycystic Kidney Disease (ADPKD), which do not show any evidence of A-IC cell presence in the cyst lining or their participation in cyst expansion. Rather, ADPKD mouse models or ADPKD patients demonstrate a cystogenic process that is entirely dependent on the expansion of principal cells [9,35].

Tsc1/Tsc2 with TCB1D7 are components of a trimolecular complex that regulates mTORC1 function by negatively regulating RHEB-GTPase, an activator of mTORC1 [4,36,37,38,39]. In the presence of mutations in *Tsc1* or *Tsc2* genes, RHEB-GTPase is no longer regulated by the TSC complex and can lead to mTORC1 hyperactivation [15,19,40]. In addition, the phosphorylation and inactivation of either Tsc1 or Tsc2 can mimic the phenotype exhibited because of mutations in *Tsc1* or *Tsc2* genes [38,41,42,43,44]. In our models, the hyperactivation of mTORC1, as evidenced by increased phospho-S6 levels, is prevalent in the cystic epithelium [9,10]. Furthermore, our results indicate that the phospho-S6 labeling in 47-day-old *Tsc1/Car2* dKO mice is significantly reduced in comparison to time-matched WT mice, suggesting that Car2 deficiency and reduced A-IC cell numbers delay cyst formation in our model of TSC renal cystic disease. While distinct cell types (principal cells vs. A-IC cells) line the epithelium of ADPKD and TSC kidney cysts [9,35], in both cases, the cystic epithelia display significant mTORC1 activation [37,45,46,47]. However, the role of mTORC1 activation in cystogenesis in ADPKD remains uncertain [48]. In humans with TSC, the inhibition of mTORC1 blunts the overgrowth of kidney cells and the development of renal tumors and cysts in TSC [10,12,14,49]. However, the discontinuation of mTOR inhibitors causes the return of TSC cysts and tumors [50]. The inhibition of mTORC1 in humans with ADPKD did not show a significant beneficial impact on kidney function and cyst volume [48]. These results display contrasting effects of mTORC1 in cystogenesis in TSC vs. ADPKD. There are no therapeutic (druggable) molecular targets to alleviate kidney cysts or tumors in mice or humans with TSC.

The transcription factor Foxi1 is indispensable in the differentiation of collecting duct A-IC cells [51,52]. The expression of *Foxi1* mRNA is significantly up-regulated in multiple mouse models of TSC [9,10]. Our published studies demonstrate that the knockout of *Foxi1* in *Tsc1* KO mice completely abrogates the development of renal cysts in these animals. Together, our studies suggest that the formation of renal cysts depends on the expansion of A-IC cells [9,10]. Previous studies indicate that *Car2* deficiency in mice leads to significant reductions in the number of A-IC cells in mouse collecting ducts [30,31,32]. Based on the above, we examined if the severity of renal cystogenesis in TSC is moderated because of the absence of Car2 and the consequent reduction in the basal number of A-IC cells. Our data indicate that in *Tsc1* KO mice, the ablation of *Car2* delays but does not prevent renal cystogenesis (Figure 1). This is illustrated by progressively increased expression of *Foxi1* and H^+^-ATPase components in the kidneys of 110-day-old vs. 47-day-old *Tsc1/Car2* dKO mice when compared to their WT and *Car2* KO counterparts. Enhanced expression of *Car12* and *Car13* in the kidneys of *Tsc1/Car2* dKO mice raises the possibility that the compensatory upregulation of these carbonic anhydrase isoforms may have overcome the inhibitory impact of *Car2* gene deletion on H^+^-ATPase function and A-IC cell proliferation.

## 4. Materials and Methods

### 4.1. Generation of Tsc1 KO, Car2 KO, Fox1 KO, Tsc1/Car2 dKO, and Tsc1/Foxi1 dKO Mice

The animal study protocols were approved by the Institutional Animal Care and Use Committees of both the University of New Mexico (protocol code 23-201353-HSC) and the Department of Veterans Affairs (protocol code 1621393-7).

All mice were housed and cared for in accordance with the Institutional Animal Care and Use Committees (IACUCs) at the University of New Mexico and Albuquerque Veterans Services. Lab personnel were IACUC-trained. Mice were housed in a 12 h light/dark cycle in temperature-controlled rooms with free access to food and water.

A variety of transgenic mice were utilized in this study. Details concerning generation and genotyping conditions for the following strains have been previously described: *Tsc1* KO [9], *Car2* KO [53], *Foxi1* KO [51], *Tsc1/Car2* dKO, *Tsc1/Foxi1* dKO [9].

### 4.2. Immunohistochemical and Immunofluorescence Microscopy

Mice were euthanized at day 47 or day 110 with an overdose of pentobarbital sodium and kidneys were harvested and placed in 4% paraformaldehyde at 4 °C for 24 h. Kidneys were then switched to 70% ethanol, paraffin-embedded, placed on slides in 5 μm sections, and processed for H&E staining.

For pS6 staining, slides were heated for 2 h at 60 °C and underwent antigen retrieval utilizing R-Universal Epitope Recovery Buffer in a 2100 Retriever (EMS; Hatfield, PA, USA). Sections were blocked for 10 min with Bloxall blocking solution (Vector Labs; Newark, CA, USA) and incubated for 20 min at room temperature in 10% normal goat serum (Vector Labs). Slides were then incubated overnight in pS6 ribosomal protein (Ser235/236) antibody (Cell Signaling; Danvers, MA, USA) at 4 °C and stained with VIP utilizing the Vectastain Elite ABC kit according to directions (Vector Labs).

Similar to above, specimens undergoing immunofluorescence were subjected to antigen retrieval and blocked in a PBS solution containing 1% BSA, 0.2% powdered skim milk, and 0.3% Triton X-100 for at least 60 min at room temperature before incubation with primary antibodies overnight at 4 °C. Afterward, slides were washed in PBS 3 × 10 min, incubated in secondary Alexa Fluor antibodies (Invitrogen; Waltham, MA, USA) for 2 h at room temperature, and cover-slipped with Vectashield mounting media (Vector Labs).

Slides were examined and images obtained with a Zeiss LSM800 utilizing Zen software (version 3.4.91.00000).

### 4.3. RNA-Seq Analysis

The RNA-seq analyses were performed by Novogene Bioinformatics Technology Co., Ltd. (Beijing, China). Briefly, total RNA was isolated from kidneys of WT, *Tsc1* KO, *Car2* KO, *Tsc1/Car2* dKO, *Foxi1* KO, and *Tsc1/Foxi1* dKO mice at 47 and 110 days of age. The isolated RNA samples were subjected to quality control analysis using an Agilent 2100 Bioanalyzer with RNA 6000 Nano Kits (Agilent, Santa Clara, CA, USA), subjected to poly A selection, fragmented, and reverse-transcribed to generate complementary DNA libraries that were utilized for sequencing analysis. Libraries were sequenced on the HiSeqTM 2500 system (Illumina, San Diego, CA, USA). Clean reads were aligned to a mouse reference genome using Hisat2 v2.0.4. Gene expression levels were determined using fragments per kilobase of transcript per million mapped fragments (FPKM) by HTSeq v0.9.1. The enrichment analysis of DET was performed using ShinyGO application (http://bioinformatics.sdstate.edu/go/, 19 February 2024).

### 4.4. Statistical Analysis

The significance of differences between the mean values +/− SD of multiple samples was examined using ANOVA. A *“p”* value of <0.05 was considered statistically significant.

## 5. Conclusions

These studies point out the importance of hyperproliferating A-IC cells that express both Tsc1 and Tsc2 proteins and therefore should have a functional TSC-RHEB-mTORC1 axis in TSC cystogenesis [9,10], a pattern that is also observed in the epithelium of renal cysts in TSC patients [11]. These studies focus on the process of renal cystogenesis in a mouse model of TSC renal cystic disease; however, additional TSC disease models that are available, such as heterozygote *Tsc2* KO (*Tsc2*^+/−^*)* and other cell-specific knockout models, need to be examined in order to confirm the role of Car2 in TSC renal cystogenesis. These studies are underway; however, due to the prolonged duration of cyst development, they could not be included in this current study. The most pertinent point of this study is the confirmation of the role of A-IC cells in TSC cystogenesis. The observation that genetic manipulations reduce the number of intercalated cells, for example, the ablation of the *Car2* gene, suggests a new approach that can complement the current mTORC1 inhibition approach that is used for the treatment of TSC renal cystic disease. The role of Car2 inhibitors in combination with mTORC1 inhibitors (Rapamycin analogs) or as a standalone therapy needs to be further examined. These studies need to be conducted in both the *Tsc1* KO as well as other TSC models, such as *Tsc2*^+/−^ mice.

## Figures and Tables

**Figure 1 ijms-25-04772-f001:**
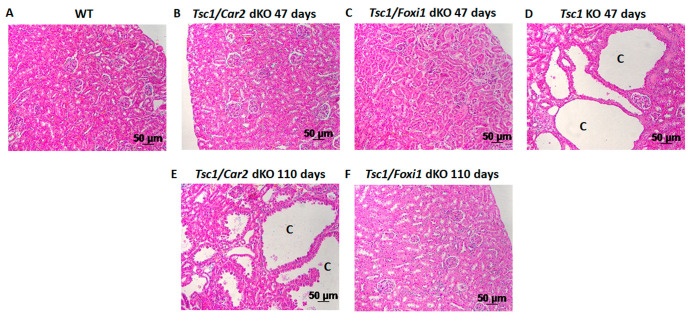
Comparison of cystogenesis and cyst progression in *Tsc1* KO and *Tsc1/Car2* dKO mice. H&E images were taken of 47 days old wild-type (WT; **A**) and *Tsc1* KO mice (**D**). These were then compared to 47- and 110-day-old *Tsc1/Car2* dKO (**B**,**E**) and *Tsc1/Foxi1* dKO (**C**,**F**) mice. Notice the reductions in the number and size of the renal cysts in 47-day-old *Tsc1/Car2* dKO compared to age-matched *Tsc1* dKO mice (**B**,**D**). images of *Tsc1/Car2* dKO mice taken at 110 days of age (**E**) show a significant increase in both cyst number and size. The life span of *Tsc1* KO mice, which is less than 60 days, did not allow for direct comparison of cyst burden to that of 110-day-old *Tsc1/Car2* dKO mice. Scale bar represents 50 μm. “C” designates cyst.

**Figure 2 ijms-25-04772-f002:**
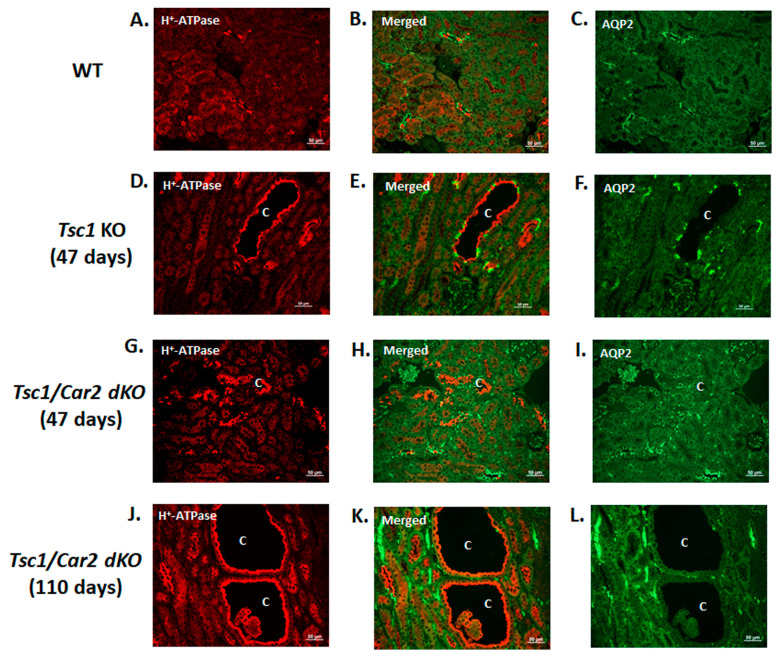
Localization of H^+^-ATPase and AQP2 in *Tsc1* KO and *Tsc1/Car2* dKO mice. Double immunofluorescence images of H^+^-ATPase (red) and AQP2 (green) were acquired form kidney sections of WT (**A**–**C**), *Tsc1* KO (**D**–**F**), 47-day-old *Tsc1/Car2* dKO (**G**–**I**), and 110-day-old *Tsc1/Car2* dKO mice (**J**–**L**). *Tsc1* KO (**D**–**F**) shows cyst formation at 47 days, while cystogenesis does not occur in *Tsc1/Car2* dKO mice until 110 days of age (**J**–**L**). However, both *TSC* mouse models show H^+^-ATPase-positive A-IC cells (red) lining the cyst epithelium, with few AQP2-positive principal cells (green). Scale bar equals 50 μm. “C” represents cysts.

**Figure 3 ijms-25-04772-f003:**
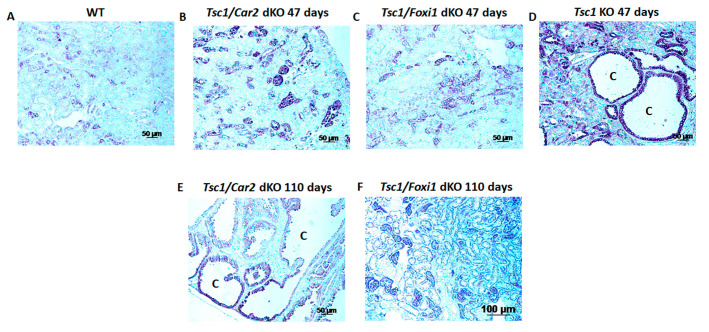
Comparison of mTORC1 activation in TSC mouse models. Immunohistochemical staining with anti-phosphorylated S6 (p-S6) antibody was used to determine the extent of mTORC1 activation. Both 47-day-old *Tsc1* KO (**D**) and 110-day-old *Tsc1/Car2* dKO mice (**E**) show extensive staining for pS6 along the luminal epithelium of the cyst. Age-matched *Tsc1/Foxi1* dKO and *Tsc1/Car2* dKO 47-day-old mice (**B**,**C**) display pS6-positive tubules but no cyst formation. The *Tsc1/Foxi1* dKO mice at 110 days did not show any signs of cyst formation (**F**). Wild-type (WT) mice were included for comparison (**A**). Scale bar equals 50 μm. “C” represents cysts.

**Figure 4 ijms-25-04772-f004:**
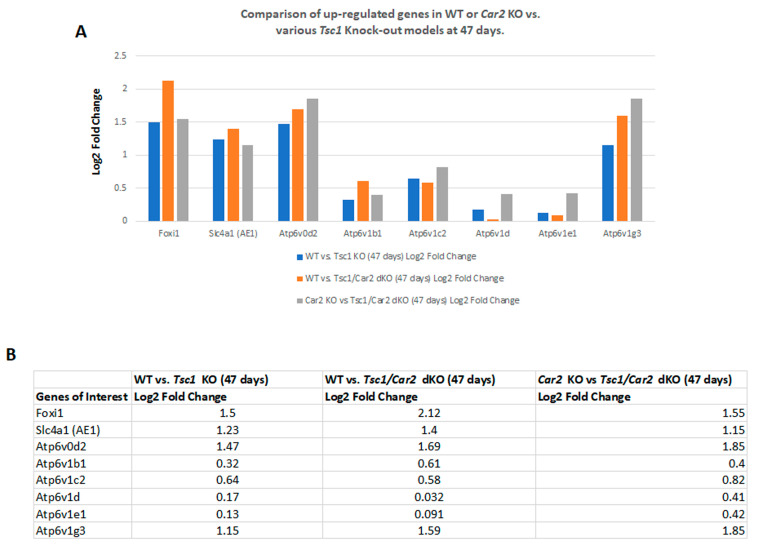
RNA-seq analysis comparing the renal transcriptomes of WT or *Car2* KO with 47-day-old mice of *Tsc1* KO or *Tsc1/Car2* dKO strains. (**A**) Graphic representation of changes in the expression of mRNAs associated with A-IC cells. Our data demonstrate that the expression of transcripts coding for *Foxi1*, *Slc4a1*, and A-IC cell-associated components of H^+^-ATPase were significantly up-regulated in the kidneys of *Tsc1/Car2* dKO mice. (**B**) Tabulated presentation of the results in part A.

**Figure 5 ijms-25-04772-f005:**
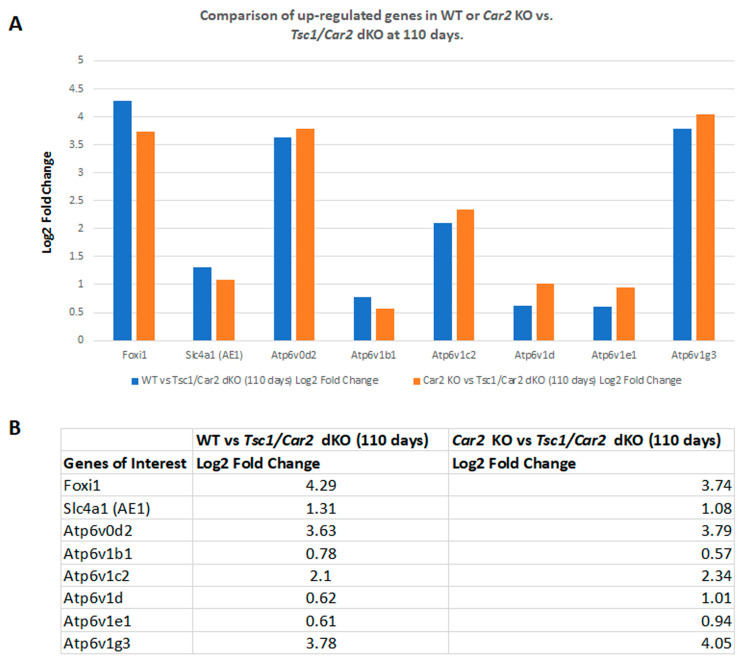
RNA-seq analysis comparing the renal transcriptomes of WT or *Car2* KO mice to those of 110-days old *Tsc1/Car2* dKO mice. (**A**) Graphic representation of changes in the expression of mRNAs associated with A-IC cells. Our data demonstrate that the expression of transcripts coding for *Foxi1*, *Slc4a1*, and A-IC cell-associated components of H^+^-ATPase were significantly up-regulated in the kidneys of *Tsc1/Car2* dKO mice. (**B**) Tabulated presentation of the results in part A. *Tsc1* KO mice were not included in this analysis due to their short life-span (less than 65 days).

**Figure 6 ijms-25-04772-f006:**
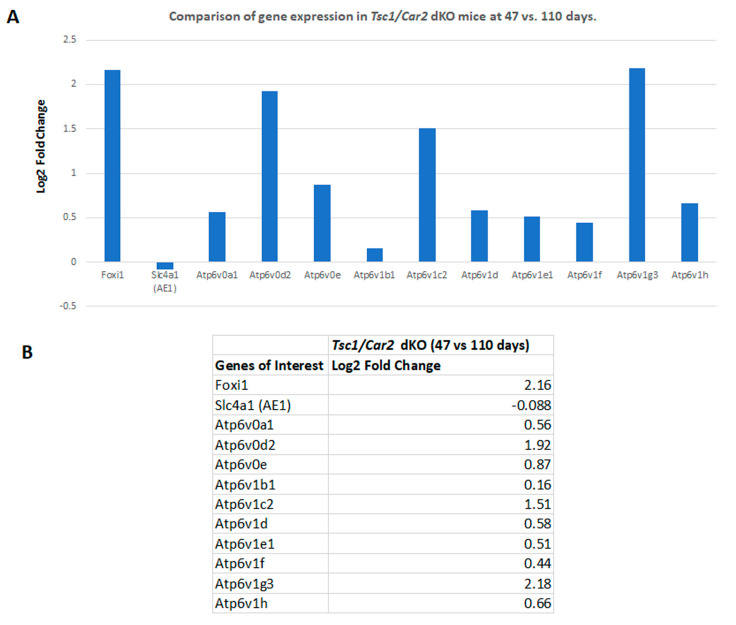
RNA-seq analysis comparing the renal transcriptomes of 47 vs. 110-day-old *Tsc1/Car2* dKO mice. (**A**) Graphic representation of changes in the expression of mRNAs associated with A-IC cells. Our data demonstrate that the expression of transcripts coding for *Foxi1*, *Slc4a1*, and A-IC cell-associated components of H^+^-ATPase were significantly up-regulated in the kidneys of *Tsc1/Car2* dKO mice. (**B**) Tabulated presentation of the results in part A.

**Figure 7 ijms-25-04772-f007:**
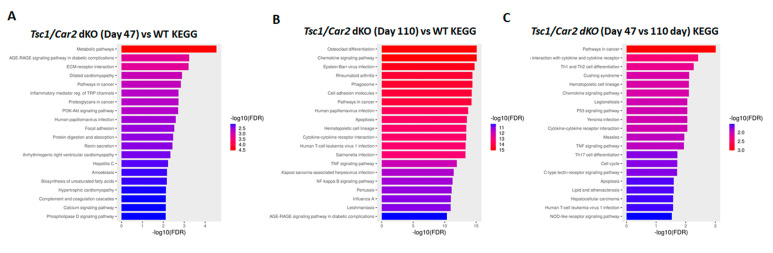
KEGG enrichment analysis of RNA-seq results. The results of RNA-seq studies were subjected to KEGG enrichment analysis. The results revealed the presence of 79, 156, and 23 enriched terms with an FDR of less than 0.05 for 47-day-old *Tsc1/Car2* dKO vs. WT mice; 110-day-old *Tsc1/Car2* dKO vs. WT mice, and 110-day-old dKO vs. 47-day-old dKO mice, respectively (for complete results of all significantly enriched terms, please refer to Appendix A).

**Figure 8 ijms-25-04772-f008:**
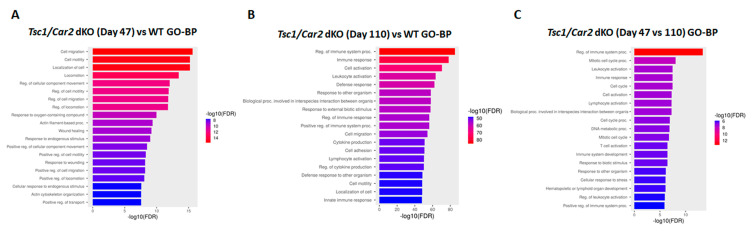
GO-BP enrichment analysis of RNA-seq results. RNA-seq results were subjected to GO-BP enrichment analysis. The outcomes revealed the presence of 939, 1000+, and 422 enriched terms that had FDRs of less than 0.05 for 47-day-old *Tsc1/Car2* dKO vs. WT mice; 110-day-old *Tsc1/Car2* dKO vs. WT mice, and 110-day-old dKO vs. 47-day-old dKO mice, respectively (for complete results of all significantly enriched terms, please refer to Appendix A).

**Figure 9 ijms-25-04772-f009:**
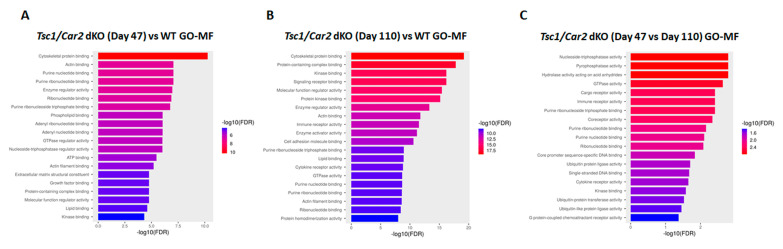
GO-MF enrichment analysis of RNA-seq results. GO-BP enrichment analysis of RNA-seq results. RNA-seq results were subjected to GO-MF enrichment analysis. These analyses revealed the presence of 128, 222, and 26 enriched terms that had FDRs of less than 0.05 for 47-day-old *Tsc1/Car2* dKO vs. WT mice, 110-day-old *Tsc1/Car2* dKO vs. WT mice, and 110-day-old dKO vs. 47-day-old dKO mice, respectively (for complete results of all significantly enriched terms, please refer to Appendix A).

## Data Availability

All datasets are included in the Appendix A and described in the text of the manuscript.

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
