# Peer review of "Carbonic Anhydrase 2 Deletion Delays the Growth of Kidney Cysts Whereas Foxi1 Deletion Completely Abrogates Cystogenesis in TSC"

_ijms, 2024, doi:10.3390/ijms25094772_

Round 1
Reviewer 1 Report
Comments and Suggestions for Authors
The manuscript titled "Carbonic Anhydrase II Deletion Delays the Growth of Kidney Cysts Whereas FOXi1 Deletion Completely Abrogates Cystogenesis in TSC" presents a comprehensive and valuable investigation of the role of Car2 and FOXi1 in kidney cystogenesis in the context of TSC. The authors used Tsc1/Car2 dKO and Tsc1/Foxi1 dKO mice models, RNA sequencing, and immunohistochemical analyses to examine the cytogenic process and the molecular pathways involved.
Comments
Given that TSC is a complex disease affecting several organs, it remains unclear how specific the observed effects of Car2 and FOXi1 deletion are to kidney cystogenesis versus other manifestations of TSC. The authors should discuss whether similar mechanisms are present in other TSC-related tumors or lesions.
Comments on the Quality of English Language
The overall quality of the english is good
Author Response
Response: The examination of the effect of Car2 deficiency in renal cystogenesis is based on two observations: 1) the renal cysts in various TSC models are primarily composed of A-intercalated (A-IC) cells; and 2) deficiency or inhibition of Car2 may significantly reduce the number of A-IC cells [Ref. 30-32]. These observations are quite unique to IC cells and renal cysts and may not apply to other TSC lesions (e.g., TSC-associated hamartomas and tubers). While the examination of other TSC associated lesions may provide very significant results and should be addressed, they are outside the purview of this manuscript.
Reviewer 2 Report
Comments and Suggestions for Authors
In this study, authors explored the role of carbonic anhydrase and FOXi in kidney cyst growth using gene knockout animal models. Their findings showed that significant cyst formation in kidney was shown in Tsc1 KO mice-47day and Tsc1/Car2 dKO mice-110day, but not shown in Tsc1/Foxi1 dKO mice-110day. In addition, the cyst burden in Tsc1/Car2 dKO mice was correlated with Foxi1 expression and mTORC1 activation. Overall, this study has merit and interest. However, there are concerns that need to be further addressed and interpreted. In parallel, the present manuscript needs a proper improvement.
1. Authors established Tsc1 KO mice, Tsc1/Car2 dKO mice, and Tsc1/Foxi1 dKO mice, and showed the different cystogenesis and cyst progression in these mouse models. However, the RNA seq and system biology analysis merely showed the differential transcriptome between WT, Tsc1/Car2 dKO mice at 47, and Tsc1/Car2 dKO mice at 110 days. Why was analysis for transcriptome of Tsc1/Foxi1 dKO mice not conducted and not further compared with the other models? Authors should well interpret it.
2. In fig.1, the rationale that why cystogenesis and cyst progression in Tsc1 KO mice-110day was not performed should be addressed. In addition, as mentioned that “Notice the reductions in the number and size of the renal cysts…”, it is suggested to show the number of cysts in each condition and the statistical analysis results for their difference.
3. In fig.3, the comparison of mTORC1 activation in the mouse models was only detected by using IHC. It is suggested to perform western blotting to determine the quantitative changes of mTORC1 activation kidney tissues in different mouse models.
4. The limitations of this study were not mentioned and discussed.
5. Northern blotting is described in methodology; however, the Northern blotting results were not shown or indicated.
6. There are typos and errors, such as #131: “50 m”.
Comments on the Quality of English LanguageThe quality of English language is fine. However, some typos and errors still need to be corrected.
Author Response
- Authors established Tsc1 KO mice, Tsc1/Car2 dKO mice, and Tsc1/Foxi1 dKO mice, and showed the different cystogenesis and cyst progression in these mouse models. However, the RNA seq and system biology analysis merely showed the differential transcriptome between WT, Tsc1/Car2 dKO mice at 47, and Tsc1/Car2 dKO mice at 110 days. Why was analysis for transcriptome of Tsc1/Foxi1 dKO mice not conducted and not further compared with the other models? Authors should well interpret it.
Response: A targeted exploration of the Tsc1/Foxi1 dKO mouse was addressed in our 2021 PNAS publication, “Kidney intercalated cells and the transcription factor FOXI1 drive cystogenesis in tuberous sclerosis complex” [Ref. 9]. As noted, the primary focus of the current paper was on the role of Car2 deficiency and the associated reduction in A-IC cells in renal cystogenesis. As such, referral to the absence of Foxi1, a transcription factor that is indispensable in IC development [Ref. 51], was to complement the data on Car2 deficiency, which reduces the number of IC cells [Refs. 30-32], and the role of A-IC cells in TSC renal cystogenesis. Future studies will delve into more detailed analysis of transcriptomes in Tsc1 KO, Tsc2 KO, and inducible principal cell-specific Tsc1 KO mice, as well as in Tsc1/Fox1 dKO mice.
- In fig.1, the rationale that why cystogenesis and cyst progression in Tsc1 KO mice-110day was not performed should be addressed. In addition, as mentioned that “Notice the reductions in the number and size of the renal cysts…”, it is suggested to show the number of cysts in each condition and the statistical analysis results for their difference.
Response: Tsc1 KO mice have a shortened lifespan of less than 60 days [Ref. 9] making it impossible to make a direct comparison of cyst burden to that of 110-day Tsc1/Car2 dKO mice. This is addressed in Lines 107-108 in the text and this explanation has now been added to the Figure 1 legend (lines 516-518). We have removed reference to cyst number to correctly reflect the conclusions regarding the cystogenesis.
- In fig.3, the comparison of mTORC1 activation in the mouse models was only detected by using IHC. It is suggested to perform western blotting to determine the quantitative changes of mTORC1 activation kidney tissues in different mouse models.
Response: The immunohistochemical (IHC) analysis of phosphorylated S6 (p-S6; indicator of mTORC1 activation) is an established method. The IHC results clearly show the differences in the activation levels of mTORC1 as indicated by the intensity of p-S6 staining; therefore, in the current study we believe that the IHC results are clear enough to obviate the need for western blot analysis.
- The limitations of this study were not mentioned and discussed.
Response: We have added additional statements in the Conclusion section (lines 242-247 and 253-254) discussing the importance of further studies using additional mouse models of TSC. Furthermore, while the knockout models examine the role of Car2 deficiency and its potential as a therapeutic target, this manuscript did not include such studies. Future studies aimed at examining the role of CAR2 inhibition or treatment with CAR2 inhibitors such as acetazolamide in conjunction with rapamycin analogs in the treatment of TSC in multiple mouse models of TSC are underway.
- Northern blotting is described in methodology; however, the Northern blotting results were not shown or indicated.
Response: Northern Blotting has been removed from the Methodology section and sub-section numbers were adjusted accordingly.
- There are typos and errors, such as #131: “50 m”.
Response: These issues have been addressed now and corrected as suggested (lines 518, 527, and 535).
Round 2
Reviewer 2 Report
Comments and Suggestions for Authors
The previous issues has been properly addressed, and the manuscript has been well improved. No further issues arose.